# Pannexins and Connexins: Their Relevance for Oocyte Developmental Competence

**DOI:** 10.3390/ijms22115918

**Published:** 2021-05-31

**Authors:** Paweł Kordowitzki, Gabriela Sokołowska, Marta Wasielak-Politowska, Agnieszka Skowronska, Mariusz T. Skowronski

**Affiliations:** 1Institute of Animal Reproduction and Food Research of Polish Academy of Sciences, Bydgoska Street 7, 10-243 Olsztyn, Poland; p.kordowitzki@umk.pl; 2Department of Basic and Preclinical Sciences, Faculty of Biological and Veterinary Sciences, Nicolaus Copernicus University, Gagarina Street 7, 87-100 Torun, Poland; 3Department of Reproduction and Gynecological Endocrinology, Medical University of Bialystok, Jana Kilińskiego Street 1, 15-089 Białystok, Poland; sokolowskagabriela@gmail.com; 4Center of Gynecology, Endocrinology and Reproductive Medicine—Artemida, Jagiellońska Street 78, 10-357 Olsztyn, Poland; m.wasielak.politowska@gmail.com; 5Department of Human Physiology and Pathophysiology, School of Medicine, Collegium Medicum, University of Warmia and Mazury, Warszawska Street 30, 10-357 Olsztyn, Poland; agnieszka.skowronska@uwm.edu.pl

**Keywords:** pannexin, connexin, oocyte, developmental competence, oogenesis, maturation, fertilization

## Abstract

The oocyte is the major determinant of embryo developmental competence in all mammalian species. Although fundamental advances have been generated in the field of reproductive medicine and assisted reproductive technologies in the past three decades, researchers and clinicians are still trying to elucidate molecular factors and pathways, which could be pivotal for the oocyte’s developmental competence. The cell-to-cell and cell-to-matrix communications are crucial not only for oocytes but also for multicellular organisms in general. This latter mentioned communication is among others possibly due to the Connexin and Pannexin families of large-pore forming channels. Pannexins belong to a protein group of ATP-release channels, therefore of high importance for the oocyte due to its requirements of high energy supply. An increasing body of studies on Pannexins provided evidence that these channels not only play a role during physiological processes of an oocyte but also during pathological circumstances which could lead to the development of diseases or infertility. Connexins are proteins that form membrane channels and gap-junctions, and more precisely, these proteins enable the exchange of some ions and molecules, and therefore they do play a fundamental role in the communication between the oocyte and accompanying cells. Herein, the role of Pannexins and Connexins for the processes of oogenesis, folliculogenesis, oocyte maturation and fertilization will be discussed and, at the end of this review, Pannexin and Connexin related pathologies and their impact on the developmental competence of oocytes will be provided.

## 1. Introduction

A breakthrough in human reproductive medicine was the birth of world’s first in vitro fertilized (IVF) baby, Louise Joy Brown, who was born in 1978. Today, there is widespread use of various assisted reproductive technologies. It is generally accepted that human oocytes have reduced developmental competence and increased aneuploidy with advancing maternal age. Therefore, the outstanding need for adequate biomarkers of oocyte developmental competence has become a high priority for research and fertility clinics around the globe since women’s first attempt at childbearing has increased in the last three decades [1]. One possible pathway, that is worth having a closer look at when intending to find such biomarkers, could be the cellular communication between the oocyte and surrounding cells inside of a follicle. The cell-to-cell and cell-to-matrix communications are crucial not only for oocytes but also for multicellular organisms in general. This latter mentioned communication is among others possible due to the Connexin and Pannexin families of large-pore forming channels. Connexin and Pannexin have a comparable 3D structure, although there is no sequence homology. Furthermore, six Connexins form a connexon and six Pannexins form a pannexon, both of which are hemichannels (Figure 1A). The Connexin family constitutes a group of homologous proteins (21 in humans), encoded by different genes [2]. The protein’s size in the members of the Connexin family is different, ranging from the smallest size of 23 kD (Connexin 23) to the largest protein size of this family at 62 kD (Connexin62) [3]. As previously mentioned Connexins are membrane channels forming gap-junctions, and more precisely, these intercellular proteins enable the exchange of some ions and molecules and therefore playing a crucial role in the communication between cells [4]. For the cellular osmoregulation, volume-regulated anion channels (VRACs) appear to be relevant. As Connexins and Pannexins, these VRACs are assembled in hexamers which are formed by the protein family named LRRC8 [5]. So far five members of the LRRCA family are described, namely LRRC8 A-E [6].

The unique gating and permeability of gap-junction channels is defined by their Connexin alignment [3,7], where channels can be (1) homomeric-homotypic, (2) homomeric-heterotypic, or (3) heteromeric-heterotypic (Figure 1B). Noteworthy, heterotypic channels are mainly built by two homomeric hexamers and only some Connexins are compatible for interrelation [8]. However, gap junction channels can also be formed by two heteromeric hexamers but so far there are no reports available which could describe heteromeric-homotypic gap-junctions in oocytes or surrounding cells, although gap junction channels can be formed containing more than one Connexin isoforms [9,10]. Interestingly, Connexins can be further divided according to their amino acid sequence homology into three groups, namely alpha Connexins and beta Connexins, whereas the third group consists of gamma, delta and epsilon Connexins [11,12,13]. Consequently, these differences in the Connexin composition imply the multi-faceted task for the physiology and developmental competence of an oocyte.

Two decades ago, Panchin and co-workers [14] discovered the Pannexin family, which constitutes a group of three glycoproteins, Pannexin 1, Pannexin 2 and Pannexin 3 [15,16]. Expression of Panx1 was confirmed in the male and female reproductive tract, but its role in reproductive cells, especially in the oocyte, still needs further elucidation. Numerous studies have been dedicated to Pannexin 1, which is encoded by the gene *PANX1* [17] and has a wide range of involvement in several physiological and pathophysiological functions [18,19]. Therefore, acquiring a deeper knowledge of Pannexin’s and Connexin’s importance for the mammalian oocyte is of high interest for the research field of reproductive medicine and for assisted reproductive technologies. This review provides an overview of current evidence on the link between oocyte developmental competence and the intercellular communication upon the Pannexin and Connexin channel proteins. Herein, their role for the processes of oogenesis, folliculogenesis, oocyte maturation and fertilization will be discussed, and at the end of this review, Pannexin and Connexin related pathologies and their impact on the oocyte’s viability and fertility will be provided.

## 2. Pannexin and Connexin Involvement in Oogenesis and Folliculogenesis

Oogenesis in mammalian species is defined as the formation and maturation of female gametes during embryogenesis and starts on the first days of the embryonic period [20]. Primordial germ cells differentiate into oogonia which proliferate to form primary oocytes. At the end of the fetal period, they begin the meiosis which is arrested at the prophase stage in many mammalian species and humans. In this state, the oocyte of women can remain even until reaching menopause. Just before each ovulation, the first meiotic division resumes [21]. Successful oogenesis requires full cooperation of oocytes and those cells surrounding them, namely granulosa and cumulus cells [21,22]. Differentiated cumulus cells are essential for oocyte nuclear and cytoplasmic maturation. They supply oocytes with the nutrients and regulatory signals needed for further development [22,23].

There are conserved homologies of up to 94% between human and murine Pannexins, and *PANX1* knockout mice have shown to be viable and fertile [17,24]. Furthermore, Pannexin 1 has been recently described to be involved in oocyte development and growth [25]. It has been shown that Pannexin 1 is localized in cumulus cells with ubiquitous expression pattern. The expression of the *PANX1* gene in bovine oocytes and cumulus cells is differential with higher expression in smaller antral follicles compared to larger antral follicles, which suggests that the expression of Pannexin 1 decreases in vivo during antral follicle development [25]. The role of Pannexin 2 and Pannexin 3 in oogenesis needs to be elucidated. The involvement of Pannexins in oocyte development is not precisely described, while the involvement of Connexins has been investigated widely.

The most well-studied Connexins in oocyte-cumulus-complexes (COCs) are Connexin 43 and 37 (Cx43 and 37, respectively) [3,16]. Both of them have a pivotal role in all stages of folliculogenesis and oocyte maturation [26,27] (Table 1). Granulosa cells express Cx43, whereas Cx37 is present in both cumulus cells and oocytes at all stages of follicular development [3,28]. Due to their differential localization (Figure 2), they are responsible for slightly different functions. Cx43 enables the communication among granulosa cells, and its lack leads to the arrest of oocyte development at primary stages. Cx37 is essential for the gap–junctional communication between oocyte and cumulus cells (Figure 2) [29,30]. Previous research provided evidence that the lack of Cx37 in female mice led to diminished follicular development at early antral stages, as well as to a reduced meiotic competence [31]. Furthermore, the deletion of the *Gja4* gene, which encodes for Cx37, resulted in a lack of gap-junctions between oocytes and cumulus cells, and additionally, in the granulosa cells, characteristics of a premature luteinization were observed [31]. Consequently, it can be assumed that Cx37 might be also involved in the signal transduction responsible for the prevention of granulosa cell luteinization prior to ovulation [3]. The knockout of Cx43 in mice was lethal, therefore it was necessary to transplant the ovaries lacking Cx43 into a kidney of another adult wild-type mouse to follow up its further postnatal development [30,32]. Loss of Cx43 led to an arrest of follicle development before antrum formation. Moreover, the absence of Cx43 impaired the follicle growth and decreased the sensitivity of granulosa cells upon growth differentiation factor 9 (GDF9) [33]. Cx43 also facilitates oogenesis by providing connexons for the plasma membranes of granulosa cells [34]. These hemichannels regulate the release of molecules and ions from cells to interact with receptors on surrounding cells. This pathway may also be supportive and crucial for folliculogenesis [34].

Interestingly, when a wild-type murine oocyte was fused with granular cells bearing a mutation of Cx43 no intercellular gap-junction could be formed, although hemichannels were present in the plasma membrane (Table 1). Additionally, folliculogenesis was diminished suggesting the pivotal role of Cx43 in forming gap-junctions in murine granulosa cells [35]. Noteworthy, the PDZ-binding domain of Cx43 may play an important role in oogenesis, and it was reported that PDZ-binding domain deletion homozygote mice rarely survive, were infertile, and their follicles were morphologically impaired [31,36].

The Follicle-stimulating hormone (FSH) secreted by the posterior lobe of the pituitary gland has its receptors among other localizations on granulosa cells [37]. FSH is required for the proper growth of more advanced follicular stages (antral follicles) which during their development become responsive to FSH [38]. It was demonstrated that FSH regulates the Cx43 protein expression (Figure 2), and its encoding mRNA (*GJA1*) expression (Figure 2) [39]. It is worth noting that the stimulation of *GJA1* mRNA abundance upon FSH in granulosa cells appears to be regulated by protein kinase A (PKA) and through the Wnt/β-catenin pathway, which suggests that FSH may upregulate steady-state levels of these mRNAs by increasing their transcription [33]. Granulosa cell communication in a cumulus-oocyte-complex (COC) is increased by FSH through the phosphorylation and translocation of Cx43 (Figure 2) [40,41]. The gene *GJA4* which encodes for Cx37 is also upregulated in oocytes upon FSH, thereby their gap-junctional communication with cumulus cells is increased [42]. Studies on the FSH dependent regulation of Connexins suggest that free Connexins form new gap-junctions once the hormone is present, and presumably the direct effect of FSH on *GJA1* expression in granulosa cells is responsible for the increase of Cx43. A recently published study revealed that the aberrant *GJA1* gene appears to provoke an arrest of follicular development in women suffering from polycystic ovary syndrome (PCOS) [22]. This female endocrine disorder may lead to infertility due to ovulatory dysfunction and polycystic ovary morphology. These data demonstrated that *GJA1* is downregulated in oocytes in women with PCOS. Another research showed impaired endothelin-1 (ET-1) expression in granulosa cells from patients with PCOS. ET-1 can promote the oocyte maturation but more research about the importance of this protein in PCOS patients is needed [43,44].

## 3. Pannexin and Connexin Involvement in Oocyte Maturation

Oocyte maturation reflects the process which is mandatory for the mammalian female gamete to be fertilized by the male gamete [21]. Oocyte maturation can be divided into four main steps, namely: (1) the nuclear maturation; (2) the cytoplasmic molecular maturation; (3) the cytoplasmic organelle maturation [45]; and (4) the epigenetic maturation [46]. In perinatal stages, oocytes of the most mammalian species are arrested at the prophase of meiosis I (prophase I). The first step of oocyte maturation contains the resumption triggered naturally by the luteinizing hormone (LH) [20]. The second step of oocyte maturation is the cytoplasmic molecular maturation where recruitment of specific transcripts for translation takes place. The aim of the third step is cytoplasmic organelles’ maturation, meaning the adequate distribution of among others cortical granules, and mitochondria, during the transition to metaphase II (MII) [21]. The epigenetic maturation step is crucial for the regulation of gene expression, nuclear architecture, and chromosome stability [20]. All in all, oocyte maturation is a precisely orchestrated process in which an undisturbed communication between the oocyte and surrounding cumulus cells as well as among granulosa cells is necessary.

Recent studies have shown that *PANX1* expression in human oocytes and eight-cell embryos is higher in comparison with cells of somatic tissues [19]. The localization of Pannexin 1, taking into consideration only localizations related to oocytes, was detected mainly on the cell membrane of human oocytes, zygotes, and at cell-cell interfaces in early embryos (Table 2) [19]. Pannexin 1 may form three variants, the no glycosylated protein (GLY0), a high-mannose glycoprotein (GLY1), and a fully mature glycoprotein (GLY2) (Figure 1A) [16]. Four independent families suffering from female infertility, in which mutations of *PANX1* were detected, were recently analyzed (Sang et al., 2019). A different *PANX1* gene mutation has been identified in each family. All four mutations (Figure 3) altered glycosylation of pannexin 1 resulting in a lack of GLY2, whereas GLY1 was maintained. Furthermore, the localization of the mutation of Pannexin 1 was in all investigated cases on the cytoplasmic side of the protein (Figure 3). This study also revealed that the cause of oocyte death phenotype was due to alternations in Pannexin 1 channel activity and led to aberrant ATP release followed by oocyte death [19]. This research suggests that changes in the degree of Panx1 channel activity may lead to oocyte death even at a stage before fertilization. These findings clearly demonstrated the pivotal role of Pannexin 1 in human oocyte development.

The Pannexin 1 inhibition with the help of an inhibitor named 10Panx was conducted during in vitro maturation of bovine COCs [25]. This study revealed that Pannexin 1 inhibition decreased cumulus cell expansion when compared to the untreated (without 10Panx) control. In the same study, it was reported that after six hours of 10Panx treatment significantly more oocytes remained at the germinal vesicle (GV) stage with significantly higher cAMP concentrations when compared to untreated counterparts, whereas after 22 h of treatment no changes in the number of oocytes reaching MII was observed [25]. This study demonstrated clearly that oocyte maturation can be delayed upon the inhibition of the Pannexin 1 hemichannel. These effects seemed to be temporary, but a higher proportion of treated oocytes reached the blastocyst stage suggesting that this delay, related to maintaining elevated cAMP levels, improves in vitro oocyte developmental competence. Finally, they demonstrated that during the maturation of oocytes with inhibited Pnx1 channels significantly fewer reactive oxygen species (ROS) were produced in comparison with the untreated control oocytes [25]. Consequently, the in vitro development of the embryo was protected against the negative effects of ROS. The role of Pannexins in oocyte maturation is crucial but not clearly defined, whereas Connexins have been well studied. An adequate cellular exchange between oocyte and follicular cells is important for oocyte maturation and development. The loss or reduction of these membrane proteins negatively affects fertility in various mammalian species [26,35,36,47,48,49,50,51]. Several Connexins were identified as playing a role in the functioning of the female reproductive tract, including Cx26, Cx32, Cx37, Cx43 and Cx45 [3,52,53]. Cumulus cells tightly surround oocytes forming a cumulus-oocyte-complex, as mentioned before. Signals from the microenvironment of the ovary have to be transferred to oocytes through communication channels. Besides providing nutrients, Connexins enable the exchange of molecules, and thereby gap-junctions are responsible for maintaining a stable pH inside the oocyte and promote chromatin structure remodeling during oocyte maturation [54]. This is mandatory for proper proliferation of the oocyte and its survival. Moreover, Connexins are required to regulate cGMP concentration in oocytes which is relevant for the meiotic arrest and later for the resumption of meiosis (Figure 2) [55].

Molecules and signals essential for molecular maturation of most mammalian oocytes are presumably transferred between cells during the first four hours of in vitro maturation (IVM) [56]. It is well documented that the meiosis arrest depends on the level of cAMP in mammalian oocytes. However, endothelin-1 (ET-1) may downregulate cAMP transfer from cumulus cells to the oocyte through Cx26, which is fundamental for the initiation of oocyte maturation [44]. Recent studies suggest that ET-1 determines oocyte maturation via endothelin receptor type B (ETRB) by downregulation Cx26 expression in cumulus cells [44]. Whole genome transcriptome microarray analyses of cumulus cells were performed. The study reported significantly lower Cx26 protein expression upon ET-1 treatment. This effect could be reversed due to a co-treatment with an ETRB antagonist [44]. The same study revealed that ET-1 affects cAMP levels in oocytes. Treatment with ET-1 significantly increased cAMP concentration in cumulus cells, whereas a decreased cAMP level in oocytes [44]. Interestingly, ET-1 did not affect Cx43 and Cx37. Moreover, Cx26 in oocytes may be involved in local cellular mechanisms during the peri-ovulation time. The expression of Cx26 is upregulated during the luteinizing hormone (LH) surge in bovine follicles but the exact mechanism is not fully understood yet [57]. After the LH surge, progesterone production is upregulated and the gap-junctions start to reduce the passage of cyclic nucleotides, which contributes to initiation of the meiotic resumption [56,58]. In porcine oocytes, it was demonstrated that Cx43 is highly controlled by gonadotropins [59,60]. Thereby, the decrease in the concentration of cGMP in oocytes induces hydrolysis of cAMP, which promotes the meiotic process and maturation of oocytes [61].

## 4. Pannexin and Connexin Involvement in Oocyte Fertilization

It was found that the interruption of gap-junctional communication within COCs is gonadotropin-dependent and decreases after germinal vesicle break-down (GVBD). The resumption of meiosis was associated with the reduction of Cx43 protein level in porcine and rat cumulus cells [58,60,62]. Analogical findings were made in bovine. The expression of Cx43 mRNA in cumulus cells was down-regulated six hours after LH release [63]. Disappearance of small Cx43-positive gap junctions, interconnecting the corona radiata cells with the oocyte, was related to GVBD as revealed by immunofluorescence and electron microscopy [64]. In humans, Cx43 gene expression was decreased in cumulus cells surrounding mature oocytes, comparing to their counterparts surrounding immature ones [65]. In case of embryonic development, there was no correlation between Cx43 expression in cumulus cells and fertilization or cleavage rate [49]. However, it was revealed that lower expression of Cx43 in cumulus cells was related to better embryo morphology on day 3 of the in vitro culture and improved blastocyst development [49,66]. Similar results were obtained with regard to Panx1. Its protein expression was decreased in cumulus cells enclosing the oocytes more competent for embryo development, when compared to the less competent oocytes [25].

In the study of Zhou and co-workers [67], removal of cumulus cells before insemination of the in vitro matured murine oocytes led to a decrease of the fertilization rate, whereas this effect could be reversed upon the supplementation of dispersed cumulus cells to the insemination medium. These results show that gap junctions are not essential during fertilization, confirming at the same time that cumulus cells can be an important source of chemotactic factors guiding the spermatozoa to the oocyte [68,69]. There are findings in *Caenorhabditis elegans*, indicating the role of innexins (gap junction proteins in invertebrates) [70], particularly innexin-14, in the sperm recruitment to the site of sperm storage [71,72]. Cumulus cells support the fertilizing ability of the spermatozoa by creating the specific microenvironment around the oocyte. Preovulatory LH surge induces the extensive production of hyaluronic acid (HA) in cumulus cells through the increase in hyaluronan synthase 2 (HAS2) expression, which leads to the deposition of a large amount of hyaluronan in their extracellular matrix and expansion [73]. It was revealed that the interaction between HA and its main surface receptor CD44 may act on tyrosine phosphorylation of Cx43—found predominantly in cumulus cells, which results in the closure of gap junctions and the activation of meiosis resumption [74].

## 5. Pannexin and Connexin Related Pathologies Which Impact Oocyte Developmental Competence

Connexins are known to play an important role in a variety of biological processes and disturbances in their functioning may cause different pathologies. Several disorders in humans were classified as related to Connexin mutations (reviewed in [75]). The most studied are mutations of Connexin 26 (Cx26) associated with hearing loss [76,77]. In this paragraph, we will concentrate on the Connexin and Pannexin linked pathologies impacting developmental competence of the female gamete. As it was described earlier in this review, gap-junctions between granulosa cells and granulosa cells and the oocyte, regulate cellular communication required for the proper oocyte maturation and ovulation. Connexin-43 and -37 seem to have predominant role in follicular development. Lack of Cx43 in mice ovaries resulted in abnormal folliculogenesis and as a consequence in developmentally incompetent oocytes with morphological anomalies such as poorly developed zona or vacuolated cytoplasm, which failed to be fertilized [26]. Similar observations were made in case of Cx37 deficiency. Targeted deletion of the Cx37 gene was associated with arrested follicle and oocyte growth and ovulation failure [31,78]. It was revealed that down-regulation of Cx43 was implicated in follicular growth arrest in women with polycystic ovary syndrome (PCOS), who are known to suffer from subfertility [22,79]. Connexin-43 gene expression was decreased in GV-stage oocytes obtained from PCOS ovaries, in relation to the oocytes recovered from the ovaries of healthy women [22,80]. Parallel findings were made regarding Cx43 protein—it was very low in oocytes obtained from PCOS patients [22]. Androstenedione dependent up-regulation of Cx43 and gap junctional communication in granulosa cells was suggested to contribute to the pathogenesis of PCOS in rat [81]. Both Cx43 protein and Cx37 mRNA expression was found to be affected in COCs of diabetic mice [82]. This could generate negative consequences such as delay in follicular growth, oocyte maturation, and higher apoptosis rate in ovarian follicles, which were observed previously as linked to diabetes [83,84]. Connexin-43 knockdown in parthenogenetically activated porcine embryos was responsible for the reduction of the membrane permeability, mitochondrial membrane potential and ATP production, and for the enhanced level of reactive oxygen species. Additionally, blastocyst developmental rate, and the total cell number were decreased [85]. The importance of Cx43 for the oocyte quality was also confirmed in bovine. High expression of Cx43 in COCs was correlated with superior developmental competence of the resulting embryos [86]. Connexins not only form gap junctions enabling intercellular communication but may also participate in transmembrane communication through formation of unopposed hemichannels. Connexons are involved in release of small metabolites and ions into the extracellular environment thus taking part in paracrine signaling [75]. However, this process has to be precisely regulated because abnormal opening of hemichannels usually leads to uncontrolled efflux of ions and metabolites, which may result in cell death. As was shown in the studies on oocyte vitrification in cat, Cx43 and Cx37 hemichannels may open during vitrification and warming, which ends with the serious injury of the oocytes [87]. In case of ovine, Cx43 mRNA was found to be also affected by vitrification and in vitro culture [88].

As mentioned previously, the main role of pannexin channels is ATP transfer from the intra- into the extracellular environment in response to activation by calcium or purinergic receptors [89]. Pannexins have been known to participate in many physiological processes such as cell differentiation [17], apoptotic cell clearance [90], initiation of inflammation [91], HIV infection [92] or neurological functions [93]. There is limited information concerning pannexin channels dysfunctions with regards to the oocyte developmental competence. In bovine, COCs Panx1 expression decreased as follicular growth progressed, that is, the protein was higher in COCs isolated from small antral follicles in comparison to its levels in large follicles. The oocytes of high developmental potential characterized with lower expression of Panx1 than their less developmentally competent counterparts. In humans, Panx1 channelopathy associated with changes in Panx1 glycosylation pattern and subcellular localization, resulted in abnormal channel activity and ATP release, and oocyte death as previously mentioned [19]. The majority of oocytes collected from these Panx1 mutated patients dedicated to intra-cytoplasmatic sperm injection (ICSI) were immature and all degenerated or did not show further development very shortly after ICSI (Figure 3B). Interestingly, the mutations showed pattern of paternal inheritance with no negative effect on male fertility, assuming that mutant Panx1 has a specific pathophysiological role during oogenesis [19].

## 6. Conclusions and Future Directions

Pannexins and Connexins act as essential channel proteins enabling the communication in oocytes to support the processes of oogenesis, folliculogenesis, maturation, fertilization, and embryonic development. As oocytes age, their number, quality, and fertilization outcomes decrease. An increasing number of studies showed that the communication as well as the exchange of molecules and ions via the Pannexin and Connexin pore-forming hemichannels and gap-junctions, are a critical determinant of oocyte developmental competence and fitness. This sparked interest within the reproductive molecular medicine field to better understand how both earlier mentioned channel proteins contribute to the determination of the competence related to oocytes’ growth, development inside of a follicle, and oocytes’ ability to be fertilized. Once the impact of Pannexins and Connexin will be fully elucidated, there will be the possibility of new technologies or treatments to modulate or restore oocyte viability. Technical advances in the three past decades have led to increasingly sophisticated assisted reproductive technologies. Nevertheless, the safe and ethical modulation of channel proteins and oocyte quality requires more comprehensive understanding of their interplay. Taken together, elucidating the molecular mechanisms that contribute to the maturation and fertilization of oocytes, the unique role of Pannexins and Connexins herein, and careful identification of therapeutic targets to improve their function and thereby oocyte health, can contribute to new strategies to enhance and prolong reproductive fitness. Importantly though, our review aimed to stimulate new research ideas on this interesting topic of cell to cell communication, which could aid in the design of novel studies and animal models.

## Figures and Tables

**Figure 1 ijms-22-05918-f001:**
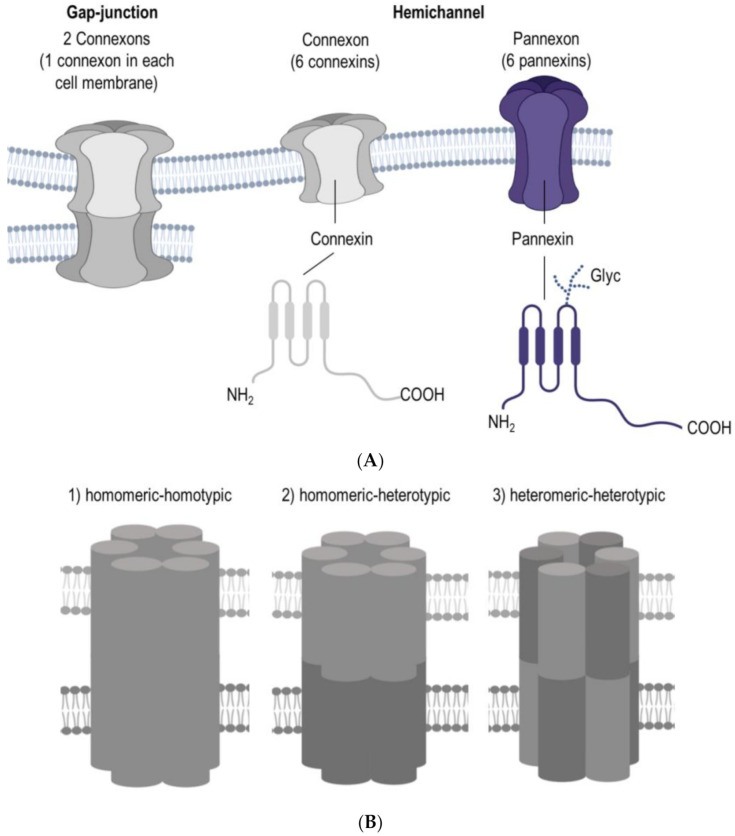
(**A**) Connexin and pannexin share a similar structure, despite the absence of sequence homology. Connexin and pannexin form functional connexon and pannexon hemichannels, re-spectively. Connexins and pannexins are transmembrane proteins with four transmembrane do-mains, two extracellular loops, one cytoplasmic loop, and cytoplasmic N- and C-terminal domains. Connexin channels can assemble into a gap junction that mediates intercellular communication, while pannexin’s extracellular loop can have different degrees of glycosylation in mammalian cells, which prevent the formation of gap junctions, three variants of glycosylation are known: GLY0, GLY1, and GLY2. (**B**) The permeability properties of gap junction channels depend on their con-nexin composition: channels can be (1) homomeric-homotypic, (2) homomeric-heterotypic, or (3) heteromeric-heterotypic. Abbreviation: Glyc = glycosylation.

**Figure 2 ijms-22-05918-f002:**
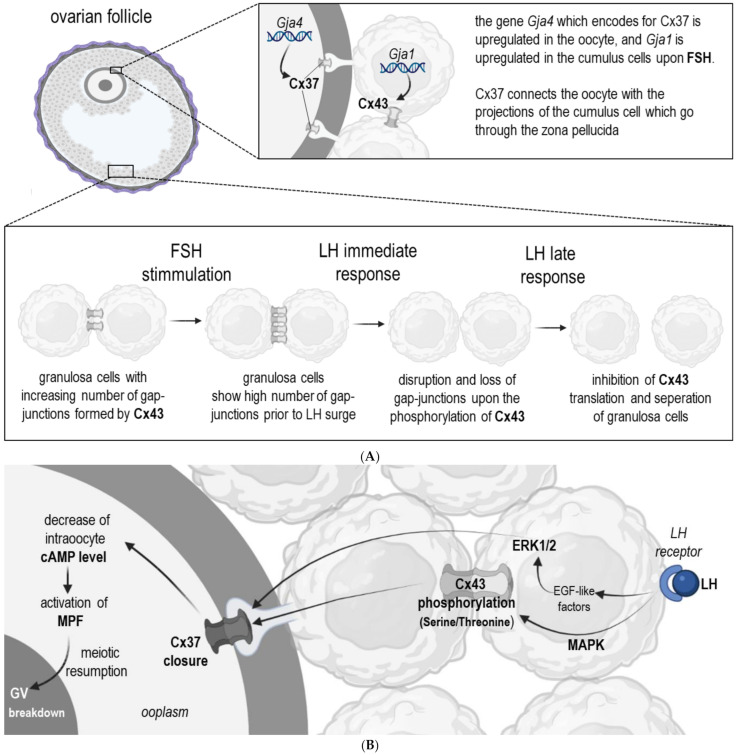
Scheme showing the regulation of gap-junctional communication (Cx43, Cx37) during the development of antral follicles prior to ovulation (**A**) and during meiotic resumption (**B**). (**A**) FSH hormone stimulates mRNA expression that codifies the Cx43/Cx37 synthesis of the gap-junctions, the amplification of functional channels, and consequently the integration to the metabolic activity. Mediated by the family mitogen-activated protein kinases, pre-ovulatory LH levels interrupt cell-cell communications by means of phosphorylation and modification of Cx43 protein conformation. This leads to interruption of the intercellular channels. The primary effect of the immediate response to LH is accompanied by elimination of the Cx43 protein, disappearance of gap-junction separation of the GCs from the oocyte. (**B**) LH stimulates the MAPK-dependent phosphorylation of Cx43, which induces the closure of Cx37and a decrease of cAMP inside the oocyte. In consequence, the maturation promoting factor (MPF) is activated and leads to resumption of meiosis and germinal vesicle (GV) breakdown. Abbreviations: Cx = Connexin, FSH = follicle stimulating Hormone, GJA = gene en-coding for connexins, LH = luteinizing hormone, ERK1/2 = extracellular signal-regulated kinases 1 and 2, MAPK = mitogen activated protein kinase, EGF = epidermal growth factor.

**Figure 3 ijms-22-05918-f003:**
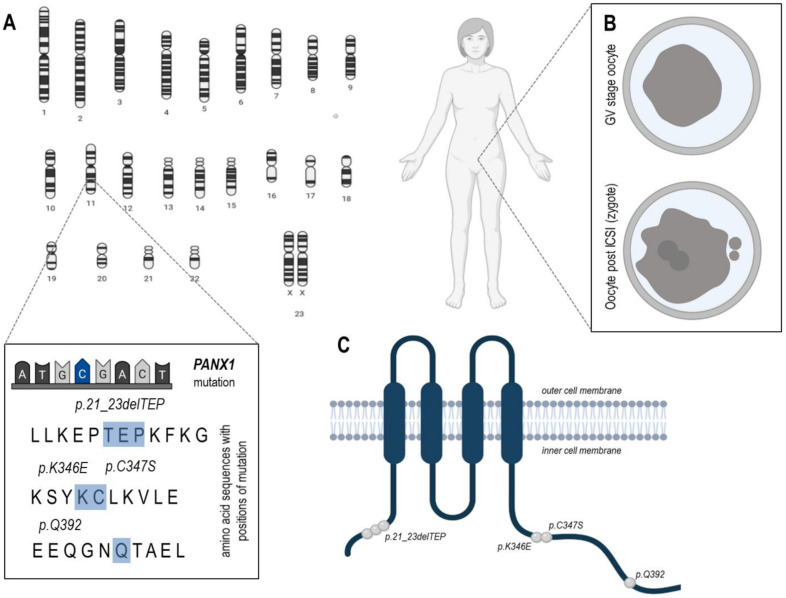
Mutation distribution of PANX1 and morphological appearance of oocytes retrieved from PANX1 channelopathy patients. (**A**) The gene Panx1 which encodes for PANX1 is located on chromosome 11 in humans. In four investigated families, four different mutations loci have been identified which led to a change in the amino acid sequence (highlighted in blue). (**B**) The upper part shows an exemplary degenerated oocyte in the germinal vesicle (GV) stage with shrunken ooplasm (dark grey), directly after oocyte retrieval; the lower part shows an exemplary degenerated zygote (fertilized oocyte) 30 h after Intra-Cytoplasmic-Sperm-Injection = ICSI, with the two pronuclei and extruded polar-bodies visible, but all specimens died at this stage. (**C**) Distribution of four disease-causing mutations in PANX1. All mutations were located in the cytoplasmic region and are highlighted as grey dots (according to [19]).

**Table 1 ijms-22-05918-t001:** Connexins involved in oocyte development and female fertility. See text for references.

Channel Protein	Encoding Gene	Reported Function in Females	Localization
Connexin 26 (Cx26)	*GJB2*	Knockout female mice died 11 days *post coitum*. Cx26 may be involved in local cellular mechanisms in oocytes among the peri-ovulation time. The expression of Cx26 is upregulated during the LH surge in the bovine follicle but the exact mechanism is not fully understood. Endothelin-1 (ET-1) may downregulate cAMP transfer from cumulus cells to oocyte via Cx26 to induce oocyte maturation. Mutations may lead to implantation failure.	Oocytes, granulosa cells, theca cells of several species.
Connexin 32 (Cx32)	*GJB1*	Knockout females remained viable and fertile.	Porcine ovary, especially theca cells. In cattle in granulosa cells of arthritic but not healthy follicles.
Connexin 37 (Cx37)	*GJA4*	Cx37 is essential for the gap-junctional communication between granulosa cells and oocyte. Cx37 localizes to gap junctions at the oocyte surface and is thereby responsible for oocyte-granulosa cell metabolic coupling. It has been shown that in mice ovaries lacking Cx37 folliculogenesis is impaired at early antral stages, as well as meiotic competence. Mutations may lead to complication with conceiving due to impaired folliculogenesis.	In murine oocytes, cumulus cells, in cumulus cells of *corona radiata*. Cx37 is present on oocytes at all stages of follicle forming.
Connexin 43 (Cx43)	*GJA1*	Cx43 provides communication among granulosa cells and its lack leads to arrest oocyte development at the primary stages. Marker potentially associated with oocyte maturation. Mutations may lead to complications with conceiving due to deficiency of oogonia and/or impaired oocyte/follicle development.	Within the follicles of numerous species. Strongly expressed in cumulus cells.
Connexin 45 (Cx45)	*GJA7*	Knockout females mice died in utero.	In pig, mouse and rat oocytes, granulosa cells and cumulus cells.
Connexin 62 (Cx62)	*GJA10*	Mice lacking CX57, a mouse orthologue of porcine CX60 and human CX62, did not exhibit any changes in oogenesis/folliculogenesis	Expressed in porcine cumulus cells and oocytes, expressed at low levels in mouse ovaries.

**Table 2 ijms-22-05918-t002:** Pannexins involved in oocyte development and female fertility. See text for references.

Channel Protein	Encoding Gene	Reported Function for Female Fertility	Localization
Pannexin 1	*PANX1*	Expression of PANX1 in bovine oocyte cumulus cells is differential with higher expression in smaller antral follicles compared to larger antral follicles. The expression of PANX1 is downregulated in vivo during folliculogenesis and oocyte maturation. PANX1 channel inhibition during in vitro maturation resulted in temporarily delayed meiotic maturation and improved in vitro developmental outcomes while decreasing intercellular reactive oxygen species. PANX1 inhibition during in vitro maturation led to maintaining elevated cAMP levels and modulation of ATP release, which delayed maturation and improved developmental competence. The mutation in PANX1 appeared to affect maturation potential in the oocytes—very few oocytes were mature, with the majority being immature and all degenerated or died very shortly after fertilization. The mutation in PANX1 led to an altered PANX1 glycosylation pattern and influenced the subcellular localization of PANX1in cultured cells. The result was the aberrant PANX1 channel activity and abnormal ATP release in oocytes. Oocytes having the mutation of PANX1, degenerated soon after retrieval due to the release of more adenosine 5′-triphosphate (ATP) to the extracellular space.	Oocytes, zygotes, early embryonic cleavage stages
Pannexin 2	*PANX2*	unknown	unknown
Pannexin 3	*PANX3*	unknown	unknown

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
