# Peer review of "Pannexins and Connexins: Their Relevance for Oocyte Developmental Competence"

_ijms, 2021, doi:10.3390/ijms22115918_

Round 1

Reviewer 1 Report

In the paper of Kordowitski and co-workers, the authors have written a comprehensive review on cell-to-cell communication within mammalian follicles involving two distinct channels (connexons and pannexons). The different expression of connexins and pannexins during oocyte and follicle development and final oocyte maturation and fertilization is summarized. The role of aberrant expression in disease is also discussed.

The review is well written and provides a thoughtful compilation of references, although could be improved to balance various relevant subject matters.  However, there are some clarifications that are required for this manuscript to be suitable for publication.

Major comments:

Please give more details about protein assembly in gap junction channels. Are there any heteromeric-homotypic type of gap-junctions?

Please would you add few comments on LRRC8 protein family?

In table 1, please, add Cx62.

Minor comments:

The authors should follow the nomenclature for genes, transcripts and proteins.

P5/l.168 – “A recently published study revealed that the Gja1 gene appears to provoke an arrest of …”  Would it be better “study revealed that aberrant Gja1 gene expression……” ?

I would recommend providing better references in: p.3/l. 97 (Bukovsky et al, 2005)

Not all references are relevant (p.3/l.103+p.4/l.106 - Liu et al, 2020; p.4/l.106 - Kordowitzki et al, 2020; p.4/l.139 - Wang et al, 2013; p.4/l.140+142 - Leybaert et al, 2003; p.4/l.156 - Grummer et al, 1999; p.4/l.159 - Yun et al, 2012; p.5/l.162 - p.5/l.164 - Chakraborty et al, 2010; p.5/l.175 - Cui et al, 2018; p.6/l.198 - Sanchez & Smitz, 2012; p.7/l.240 - Berisha et al, 2009; Nitta et al, 2010; p.7/l. 248 - Wigglesworth et al, 2013; p.8/l.276 - Shinohara et al, 2018)

Please, check the references:

p.5/l.162 - Johnson et al, 2002 – (probably year 1999?)... but Johnson appears twice in reference list 49+50)

p.5/l.169 – Liu et al. is probably 2016

p.6/l.191 - add reference for (3) the cytoplasmic organelle maturation, and (4) the epigenetic maturation

p.7/l.239 - Tong et al, 2007…probably 2006

p8/l.272 - Shimada et al, 2003 and p.9/l.291 - Shimada et al, 2001… but only one reference in the list

p11/l.360 - Dufrense & Cyr, 2014  … Dufresne is correct

p11/l.364 - Lee, 2014 is not in the reference list

I suggest being consistent in formatting the Reference list.

Please compare all references found in the Reference list with those in text body (28, 48, 50, 75, 90, 95, 111, 116).

Reference 74 is not complete.

Author Response

MS ID: ijms-1201291

Dear Joanna Banot and Reviewers,

Thank you for inviting us to respond to the very thoughtful and constructive reviewer comments. We greatly appreciate the reviewers time and believe our revised review has become more well-rounded as a result.

We have incorporated all suggestions throughout the manuscript. Below is a point-by-point response to reviewers’ comments to clarify which edits were made.

We are happy to respond to additional requests if they arise.

Sincerely,

Mariusz Skowronski

Please note our following explanations:

Detailed answers to reviewer 1

REVIEWER: “Please give more details about protein assembly in gap junction channels. Are there any heteromeric-homotypic type of gap-junctions?”

ANSWER: We want to thank the reviewer for their thorough review of our manuscript.  This is a very important point and in response, we have added the following sentences: Noteworthy, heterotypic channels are mainly built by two homomeric hexamers and only some Connexins are compatible to interrelate (Koval et al, 2014). However, gap junction channels can also be formed by two heteromeric hexamers but so far there are no reports available which could describe heteromeric-homotypic gap-junctions in oocytes or surrounding cells, although gap junction channels can be formed containing more than one Connexin isoforms (Cottrell GT & Burt,2005; Koval, 2006). Interestingly, Connexins can be further divided according to their amino acid sequence homology into three groups, namely alpha Connexins and beta Connexins, whereas the third group consists of gamma, delta, and epsilon Connexins (Abascal & Zardoya, 2013; Sohl & Willecke, 2004; Beyer & Berthoud;2009).

REVIEWER: “Please would you add few comments on LRRC8 protein family?”

ANSWER: We thank the reviewer for his/her erudite comment, which we appreciate. In response, we now acknowledge this important point, and we have added the following sentences: For the cellular osmoregulation, volume-regulated anion channels (VRACs) appear to be relevant. As Connexins and Pannexins, these VRACs are assembled in hexamers which are formed by the protein family named LRRC8 (Jentsch, 2016). So far five members of the LRRCA family are described, namely LRRC8 A-E (Voss et al, 2014).

REVIEWER: “In table 1, please, add Cx62.”

ANSWER: We want to thank the reviewer for this suggestion. In response, we have added the Connexin 62 to the table (pXX):

Channel protein

Encoding gene

Reported function in females

Localization

Connexin 62 (Cx62)

GJA10

Mice lacking CX57, a mouse orthologue of porcine CX60 and human CX62,did not exhibit any changes In oogenesis/folliculogenesis

Expressed in porcine cumulus cells and oocytes,

expressed at low levels in mouse ovaries

REVIEWER: The authors should follow the nomenclature for genes, transcripts and proteins.

ANSWER: We want to thank the reviewer for this suggestion. In response, we have changed the nomenclature of genes from Gja1 (and others) to GJA1 and the respective proteins are written as Connexin and Pannexin.

REVIEWER: P5/l.168 – “A recently published study revealed that the Gja1 gene appears to provoke an arrest of …”  Would it be better “study revealed that aberrant Gja1 gene expression……” ?

ANSWER: We want to thank the reviewer for this suggestion. In response, we have changed the sentence as following: “A recently published study revealed that aberrant GJA1 gene expression…”.

REVIEWER:  I would recommend providing better references in: p.3/l. 97 (Bukovsky et al, 2005).

ANSWER: We want to thank the reviewer for this suggestion. In response, we have replaced Bukovsky et al, 2005 with: Schall PZ, Latham KE. Essential shared and species-specific features of mammalian oocyte maturation-associated transcriptome changes impacting oocyte physiology. Am J Physiol Cell Physiol. 2021 Apr 21. doi: 10.1152/ajpcell.00105.2021.

REVIEWER:  Not all references are relevant (p.3/l.103+p.4/l.106 - Liu et al, 2020; p.4/l.106 - Kordowitzki et al, 2020; p.4/l.139 - Wang et al, 2013; p.4/l.140+142 - Leybaert et al, 2003; p.4/l.156 - Grummer et al, 1999; p.4/l.159 - Yun et al, 2012;p.5/l.162 - p.5/l.164 - Chakraborty et al, 2010; p.5/l.175 - Cui et al, 2018; p.6/l.198 - Sanchez & Smitz, 2012; p.7/l.240 - Berisha et al, 2009; Nitta et al, 2010; p.7/l. 248 - Wigglesworth et al, 2013; p.8/l.276 - Shinohara et al, 2018)

ANSWER: We want to thank the reviewer for this suggestion. We agree with the reviewer that some references are not relevant, and we have carefully checked all the named citations.  In response, we decided to delete the following ones: Grummer et al, 1999; Yun et al, 2012; Chakraborty et al, 2010; Nitta et al, 2010; furthermore, we have deleted all references related to the male reproductive tract, since Reviewer 2 asked to do so.  

REVIEWER:  Please, check the references:

p.5/l.162 - Johnson et al, 2002 – (probably year 1999?)... but Johnson appears twice in reference list 49+50)-

ANSWER: We would like to clarify that there are two relevant references Johnson et al, 2002 and 1999. The latter mentioned was missing in the text, therefore, we have added this citation in the revised manuscript (p. 4).

REVIEWER:  p.5/l.169 – Liu et al. is probably 2016 

ANSWER: We would like to clarify that there are two relevant references Liu et al 2016 and 2020 (p.13)

REVIEWER:  p.7/l.239 - Tong et al, 2007…probably 2006

ANSWER: We would like to clarify that there are two relevant references Tong et al 2006 and 2007 (p.4 and (p.8).

REVIEWER:  p8/l.272 - Shimada et al, 2003 and p.9/l.291 - Shimada et al, 2001… but only one reference in the list

ANSWER: We would like to clarify that Shimada et al 2003 was deleted, whereas Shimada et al, 2001 remained.

REVIEWER:  p11/l.360 - Dufrense & Cyr, 2014  … Dufresne is correct

ANSWER: We would like to clarify that the citation has been deleted since it refers to the male reproductive tract which reviewer 2 wished to be deleted.

REVIEWER:  p11/l.364 - Lee, 2014 is not in the reference list

ANSWER: We would like to clarify that the citation has been deleted since it refers to the male reproductive tract which reviewer 2 wished to be deleted.

REVIEWER:  p.6/l.191 - add reference for (3) the cytoplasmic organelle maturation, and (4) the epigenetic maturation.

ANSWER: We want to thank the reviewer for this suggestion. In response, we have add the following references for cytoplasmic organelle maturation: (Trebichalská Z, Kyjovská D, Kloudová S, Otevřel P, Hampl A, Holubcová Z. Cytoplasmic maturation in human oocytes: an ultrastructural study †. Biol Reprod. 2021 Jan 4;104(1):106-116. doi: 10.1093/biolre/ioaa174. (p.7)

and for and (4) the epigenetic maturation:

He M, Zhang T, Yang Y, Wang C. Mechanisms of Oocyte Maturation and Related Epigenetic Regulation. Front Cell Dev Biol. 2021 Mar 19;9:654028. doi: 10.3389/fcell.2021.654028 (p.7)

REVIEWER:  I suggest being consistent in formatting the Reference list.

ANSWER: We want to thank the reviewer for this suggestion. We have improved the reference list.

REVIEWER:  Please compare all references found in the Reference list with those in text body (28, 48, 50, 75, 90, 95, 111, 116).

ANSWER: We want to thank the reviewer for this comment. We would like to clarify that some references did not appear in the text body since they were related to the statements in the table. In the revised manuscript we have addressed this comment and now all references appear in the text body.

REVIEWER:  Reference 74 is not complete.

ANSWER: We want to thank the reviewer for this comment. This reference referred to the male reproductive tract and therefore the reference was deleted according to the suggestion of reviewer 2.

Reviewer 2 Report

The manuscript "Pannexins and Connexins: Their relevance for oocyte developmental competence" is a review article discussing the role of these channel proteins in oocyte development and maturation.

This is a well written manuscript that does require some modification prior to publication.

authors state that oocytes lose developmental competence and fertilization ablility as they age. (page 1 lines 45-46.).  This statement should be modified to indicate that oocyte have reduced developmental competence and increased aneuploidy as they age (fert potential isnt' really reduced and 10yr before menopause isn't an accepted reference time period).

Several points/citation are repeated throughout the text. These should be edited to avoid the repeated statement. This can also help reduce the length of the manuscript.

The section on fertilization should be greatly reduced. Most of the paragraph talks about sperm and spermatogensis, which is not the focus of the paper. This can be removed.  It is known that cumulus cells are not required for fertilization (ICSI is performed during IVF routinely).  So, a short paragrpah should be feasible.

A figure should be added about connections and pannexins and phosphorylation and the impact on meiotic resumption.  This can reflect the impact of LH on phosphorylation of connections and the subsequent impact on cAMP levels inside the egg and meiotic resumption.

There are several grammatical errrors that should be fixed.

Paragraphs should be divided up to improve readability...page 7 line236 can be a new paragraph, page 4 line 116, page 2 line 55.

The paragraph on pathologies restates several points and dicusses knockouts..these can be removed to help shorten and improve the focus of the manuscript.   This section should focus on the pathology (PCOS as mentioned).  Do authors know if any deaf patients mentioned as a result of the connexin issues also display infertility?

In the conclusion, how practically could connexins or pannexins be used to treat infertility. This doesn't seem very likely.  We know clinically that we can mature eggs and fertilize (ICSI) without cumulus cells.  it seems these channels are most importatn for development in vivo.  It is unclear how there may be a therapy in vitro unless manipulation of the channels could someone be used to improve oocyte maturation in specific patient populations. Perhaps discuss use of IVM as a therapy and use chemicals to manipulate connexins to delay meiosis to improve IVM (an accepted approach that many use already by altering CAMP levels to permit cytoplasmic maturation before letting nuclear maturation proceed).

Author Response

MS ID: ijms-1201291

Dear Joanna Banot and Reviewers,

Thank you for inviting us to respond to the very thoughtful and constructive reviewer comments. We greatly appreciate the reviewers time and believe our revised review has become more well-rounded as a result.

We have incorporated all suggestions throughout the manuscript. Below is a point-by-point response to reviewers’ comments to clarify which edits were made.

We are happy to respond to additional requests if they arise.

Sincerely,

Mariusz Skowronski

Please note our following explanations:

Detailed answers to reviewer 2

REVIEWER: “authors state that oocytes lose developmental competence and fertilization ablility as they age. (page 1 lines 45-46.).  This statement should be modified to indicate that oocyte have reduced developmental competence and increased aneuploidy as they age (fert potential isnt' really reduced and 10yr before menopause isn't an accepted reference time period).

ANSWER: We appreciate the suggestion. In response, we have improved the sentence as following: It is generally accepted that human oocytes have reduced developmental competence and increased aneuploidy with advancing maternal age.

REVIEWER: “Several points/citation are repeated throughout the text. These should be edited to avoid the repeated statement. This can also help reduce the length of the manuscript.

ANSWER: We thank the reviewer for his/her query and comment. We agree, and we apologize that we have failed to delete repetitions in the previous version. In response we deleted statements which appeared twice or more times.

REVIEWER: “The section on fertilization should be greatly reduced. Most of the paragraph talks about sperm and spermatogensis, which is not the focus of the paper. This can be removed.  It is known that cumulus cells are not required for fertilization (ICSI is performed during IVF routinely).  So, a short paragrpah should be feasible.”

ANSWER: We thank the reviewer for his/her query and comment. We agree, that the statements related to the male reproduction can be removed. In response, the fragment concerning Cx and Pnx role in spermatogenesis was completely removed. Moreover, the part related to the role of cumulus cells in fertilization process was also shortened. At present, the paragraph is brief and we hope it will met the expectations of the Reviewer.

REVIEWER: “A figure should be added about connections and pannexins and phosphorylation and the impact on meiotic resumption.  This can reflect the impact of LH on phosphorylation of connections and the subsequent impact on cAMP levels inside the egg and meiotic resumption.”

ANSWER: We thank the reviewer for his/her query and comment. In response, we have added the following figure:

Figure 2(B): LH stimulates the MAPK-dependent phosphorylation of Cx43 what induces the closure of Cx37and a decrease of cAMP inside the oocyte. In consequence, the maturation promoting factor (MPF) is activated and leads to resumption of meiosis and germinal vesicle (GV) breakdown.

REVIEWER: “There are several grammatical errrors that should be fixed.”

ANSWER: We have improved the manuscript using the program “Grammarly” and it has been edited further by a native English speaker.

REVIEWER: “Paragraphs should be divided up to improve readability...page 7 line236 can be a new paragraph, page 4 line 116, page 2 line 55.”

ANSWER: We thank the reviewer for this useful suggestion. In response, we have added several paragraphs throughout the entire manuscript.

REVIEWER: “The paragraph on pathologies restates several points and dicusses knockouts..these can be removed to help shorten and improve the focus of the manuscript. This section should focus on the pathology (PCOS as mentioned).  Do authors know if any deaf patients mentioned as a result of the connexin issues also display infertility?”

ANSWER: We thank the reviewer for this useful suggestion. In response, we have reduced the number of statements regarding K.O., some of them remained since they are important to underline the role of a given gene, respectively. Besides, we would like to clarify that a mutation of the gene encoding for Cx26 leads to non-syndromic hearing loss and deafness (DFNB1). According to the literature, there are no information about infertility in those patients. In patients suffering from Deafness-Infertility Syndrome, male infertility was reported, however, in this disease, mutations of the genes CATSPER2 and STRC are responsible. Women with the same mutation are fertile. Please note the following references:

https://molecularcytogenetics.biomedcentral.com/articles/10.1186/1755-8166-6-19 https://philarchive.org/archive/LEEDAP                                         https://link.springer.com/article/10.1007/s12070-014-0711-9 prenatal testing

REVIEWER: “In the conclusion, how practically could connexins or pannexins be used to treat infertility. This doesn't seem very likely.  We know clinically that we can mature eggs and fertilize (ICSI) without cumulus cells.  it seems these channels are most importatn for development in vivo.  It is unclear how there may be a therapy in vitro unless manipulation of the channels could someone be used to improve oocyte maturation in specific patient populations. Perhaps discuss use of IVM as a therapy and use chemicals to manipulate connexins to delay meiosis to improve IVM (an accepted approach that many use already by altering CAMP levels to permit cytoplasmic maturation before letting nuclear maturation proceed).”

ANSWER: We thank the reviewer for this comment. We do also have experience in using cAMP modulators for bovine in vitro embryo production but unfortunately there are no reports available which show a modulation of Cx or Pnx activity by supplementing in vitro media with a compound, and there is no available literature on modifications of Cx and Pnx channels expected to improve efficiency of human IVF/infertility treatment.
